# Research on Estimation Method of Geometric Features of Structured Negative Obstacle Based on Single-Frame 3D Laser Point Cloud

**Xingdong Li [1]**, **Zhiming Gao [1]**, **Xiandong Chen [1]**, **Shufa Sun [2,\*]** and **Jiuqing Liu [1,\*]**

[1] College of Mechanical and Electrical Engineering, Northeast Forestry University, Harbin 150040, China; lixd@nefu.edu.cn (X.L.); 2020115690@nefu.edu.cn (Z.G.); zhenzhidemaoyi@nefu.edu.cn (X.C.)

[2] College of Engineering and Technology, Northeast Forestry University, Harbin 150040, China

\* Correspondence: ssfangel@163.com (S.S.); nefujdljq@163.com (J.L.); Tel.: +86-198-4615-1325 (S.S.); +86-173-8286-0481 (J.L.)

**Abstract:** A single VLP-16 LiDAR estimation method based on a single-frame 3D laser point cloud is proposed to address the problem of estimating negative obstacles' geometrical features in structured environments. Firstly, a distance measurement method is developed to determine the estimation range of the negative obstacle, which can be used to verify the accuracy of distance estimation. Secondly, the 3D point cloud of a negative obstacle is transformed into a 2D elevation raster image, making the detection and estimation of negative obstacles more intuitive and accurate. Thirdly, we compare the effects of a StatisticalOutlierRemoval filter, RadiusOutlier removal, and Conditional removal on 3D point clouds, and the effects of a Gauss filter, Median filter, and Aver filter on 2D image denoising, and design a flowchart for point cloud and image noise reduction and denoising. Finally, a geometrical feature estimation method is proposed based on the elevation raster image. The negative obstacle image in the raster is used as an auxiliary line, and the number of pixels is derived from the OpenCV-based Progressive Probabilistic Hough Transform to estimate the geometrical features of the negative obstacle based on the raster size. The experimental results show that the algorithm has high accuracy in estimating the geometric characteristics of negative obstacles on structured roads and has a practical application value for LiDAR environment perception research.

**Keywords:** LiDAR; negative obstacle detection; single frame point cloud; point cloud projection; point cloud denoising; geometric feature estimation

## 1. Introduction

Ambient sensing is the critical technology for ALV (Autonomous Land Vehicles) /UGV (Unmanned Ground Vehicles) to achieve autonomous navigation in outdoor environments. Negative obstacles such as ditches, trenches, potholes, puddles, and steep hills in unstructured environments seriously affect the safe driving of ALV/UGV. Therefore, accurate negative obstacle detection is of particular importance in the field of unmanned driving. Estimating a negative obstacle's geometry at a given distance is still challenging because the obstacle is located below the ground, which is difficult to detect with vehicle sensors.

Tingbo Hu et al. [1] proposed an image sequence-based negative obstacle detection algorithm for negative obstacle detection. Their algorithm is based on the phenomenon that a negative obstacle is 'darker' than the surrounding badlands and that the darker the distance, the more pronounced it is. Also, different cues are combined in a Bayesian framework to detect obstacles in the image sequence. L. Matthies et al. [2] proposed a negative obstacle detection method based on infrared features. The method is based on the phenomenon that negative obstacles tend to dissipate less heat and are warmer than the surrounding terrain at night, and performs local intensity analysis of the infrared images to mark areas of significant intensity as potential negative obstacles [3]. The final negative

obstacle validation is then performed by multi-frame verification and fusion. Arturo L. Rankin et al. [4] further coupled nighttime negative obstacle detection with thermic feature-based cues and geometric cues based on stereo distance data, using edge detection to generate closed contour candidate negative obstacle regions and geometrical filtering, to determine whether they are located in the ground plane. Negative obstacle cues were fused from thermic features, geometry-based distance image analysis, and geometry-based topographic map analysis.

These three typical methods for detecting negative obstacles have some limitations in temperature or lighting requirements, and are not sufficiently robust [5] to detect negative obstacles. Image sequence-based methods have the phenomenon of misreporting shadows of vegetation on the ground as negative obstacles; the most significant limitation of the thermal infrared image-based method is that it can only detect negative obstacles at night and greatly influence the climate and environment. Compared with infrared and visual sensors, LiDAR has the advantage of directly and accurately capturing the distance information of objects without being affected by conditions such as light and weather, so it is widely used in ALV/UGV [6] environmental perception technology.

LiDAR has a superior position and role in the field of negative obstacle detection. It has the advantages of high lateral resolution, high range detection accuracy, and strong anti-active interference ability. A more accurate detection method uses HDL-64 LiDAR or VLS-128 LiDAR on large vehicles, but the higher price of multi-beam LiDAR makes it difficult to popularize. Single-beam LiDAR with a rotating mechanical structure is frequently used on small and micro vehicles. Shang E et al. [7] proposed a negative obstacle detection method based on dual HDL-32 LiDAR with a unique dual LiDAR mounting method and a feature fusion based on the AMFA (adaptive matching filter based algorithm) algorithm. The FFA (feature fusion-based algorithm) algorithm fuses all features generated by different LiDAR or captured in different frames. The weight of each feature is estimated by the Bayes rule. The algorithm has good robustness and stability, a 20% increase in detection range, and a reduction in calculation time by two orders of magnitude compared to state-of-the-art technology. Liu Jiayin et al. [8] proposed an environment sensing method based on dual HDL-32 LiDAR, which can significantly improve the vehicle forward to LiDAR spot density, compared to the simple HDL-64 LiDAR spot density through a unique LiDAR mounting method. Wang Pei et al. [9] proposed a negative obstacle detection algorithm with single-line LiDAR and visionary fusion. The method lacks in the accurate estimation of negative obstacle geometric features, and the detection range is small and the accuracy insufficient. The literature [10] uses a Kinect sensor to detect negative obstacles and convert them into the laser scan data. The literature [11] proposes a set of algorithms for general obstacle feature extraction using radar and images, and a contour extraction method using multilayer techniques specifically applicable to negative obstacle detection. The literature [12] uses stereo information in combination with saliency to initialize the energy function, and uses color information to optimize the results. Then, the optimization results are hysteresis thresholds to reach the final negative obstacle region. All of the above methods can achieve better environmental sensing and save hardware costs compared to a single multi-beam LiDAR, but the accuracy of the detection methods varies and the combination of LiDAR methods still fails to completely solve the expensive cost problem.

This paper proposes a single multi-line radar method to improve the accuracy of negative obstacle geometry feature estimation with lower hardware cost to address the above problems. The specific contributions of the method are as follows:

(1) Since it is difficult to measure feature lengths in 3D point cloud data, this paper proposes to convert 3D point cloud data into 2D elevation raster maps to estimate geometric features, which reduces the difficulty of estimation;

(2) Selection of the most suitable filter for denoising negative obstacle point cloud data and denoising elevation images among many 3D and 2D denoising methods, and proposal of a denoising system applicable to 3D to 2D maps;

(3)   A proposed method for estimating geometric features based on two-dimensional elevation negative obstacle images;

(4)   A method for detecting the horizontal distance from negative obstacles to LiDAR is proposed.

The method is simple, computationally convenient, and low-cost, breaking the previous costly methods such as dual multi-beam LiDAR and joint calibration of LiDAR and camera or combination of IMU [13] inertial guidance and LiDAR, and using a single VLP-16 LiDAR to accurately estimate the geometry of negative obstacles on the horizontal ground by using virtual images generated from the data and some geometric operations. This method saves hardware costs and frees up more space and money to implement driverless technology.

## 2. Data Preprocessing

After calibrating external parameters of the LiDAR [14–16], the original point cloud was collected, and then a PassThrough filter [17] was applied to detect the negative obstacle. Then, the denoising effects of the StatisticalOutlierRemoval filter [18], RadiusOutlier removal, and Conditional removal [19] were compared. The StatisticalOutlierRemoval filter processed the negative obstacle point cloud and achieved an excellent denoising effect. The 3D negative obstacle point cloud was projected onto the XY plane to obtain a 2.5D negative obstacle elevation image, and 0.02 m and 0. 05 m rasters (rasterization) were added to the elevation image to obtain the elevation raster image. The denoising effects of the Gauss filter [20], Median filter [21], and Mean filter [22] were compared, and the Median filter that best met the denoising requirements of this experiment was used. At this point, it was determined if the elevation raster image was the first frame of data that hit the front wall of the negative obstacle precisely and, if so, we continued to estimate the geometric features; if not, the negative obstacle point cloud needed to be selected again. Based on the elevation raster map, a distance measurement method was built to estimate the length and width of the negative obstacle on the image, and then the geometric features of the negative obstacle were calculated by multiplying the product of the rasterized dimensions and the number of pixels. The flowchart of the single-frame 3D laser point cloud-based structured negative obstacle geometry estimation is shown in Figure 1.

### 2.1. Passthrough Filter to Locate Point Clouds at Negative Obstacles

To precisely locate the negative obstacle point cloud location, we needed to use the PassThrough filter in PCL (Point Cloud Learning) to filter out the point cloud data other than the negative obstacle to facilitate our experiment's continuation [23].

The original point cloud data image and the rendering after the PassThrough filter are shown in Figures 2 and 3, respectively. As can be seen, many point clouds in Figure 2 records the raw point cloud data for negative obstacle detection in both environments, where (a)/(d) is the raw point cloud *x-y* view in both environments, (b)/(e) is the raw point cloud *y-z* view in both environments, and (c)/(f) is the raw point cloud *x-z* view in both environments.

More sharp noise and detail was due to the ambient surface's prismatic structure; as a result, we used the PassThrough filter in PCL to filter the original point cloud data, i.e., set a channel based on the original point cloud spatial coordinate system that only belongs to the negative obstacles, and filtered out the points outside the range of the channel to keep the point cloud inside the channel. This "channel" in PCL is specifically expressed as a limited range set for a certain axis, Points outside the range in the Z-axis direction are filtered out by PassThrough filter (in the case of a certain distance between the background and the foreground, you can get rid of the background), the filtering effect depends on the data and filtering parameters.

Figure 3 shows comparison before and after PassThrough filter processing, where (a) is before processing and (b) is after processing. It can be seen that the other point cloud

data in (b) are basically cleared, there is some outlier noise near the negative obstacle, and the negative obstacle point cloud is kept better.

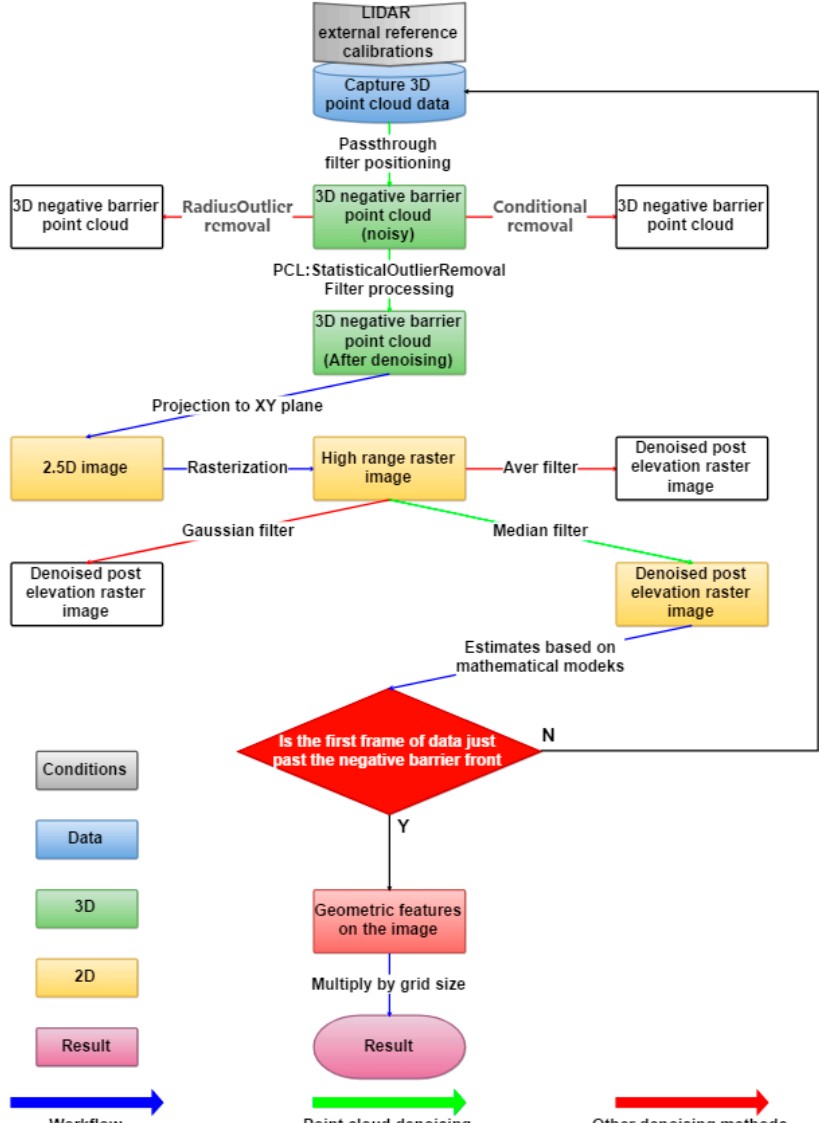

**Figure 1.** Flow chart for estimating geometric features of structured negative obstacles based on a single-frame 3D laser point cloud.

### 2.2. Point Cloud Denoising

In the process of scanning and measuring the surface of an object based on LiDAR, the 3D point cloud data must be denoised and smoothed prior to data processing for these reasons which are the external vibrations of the LiDAR, roughness of the detected negative obstacle surface, mirror reflection of the LiDAR and other artificial or random factors.

Outliers exist in the original data due to problems such as occlusion, which will affect the estimating accuracy. Our method compares three commonly used filters to remove centrifugal points, namely, the StatisticalOutlierRemoval filter, RadiusOutlier removal, and Conditional removal. Where the StatisticalOutlierRemoval filter performs statistical analysis of each point in the point cloud, it defines the number of search points, is used to calculate the distance mean, and filters outliers by making statistical judgments about the distance between the query points and the set of neighboring points. The RadiusOutlier removal is used to set a search radius and determine the number of neighboring points within the set radius, as shown in Figure 4. If the radius is set and the threshold range

is 1, then only the yellow target is removed, if the threshold range is 2, then both yellow and green range points are removed. Conditional removal, on the other hand, is more versatile and is equivalent to defining a filtered condition that removes specific points from the target point cloud. Removal can define more complex environmental filtering.

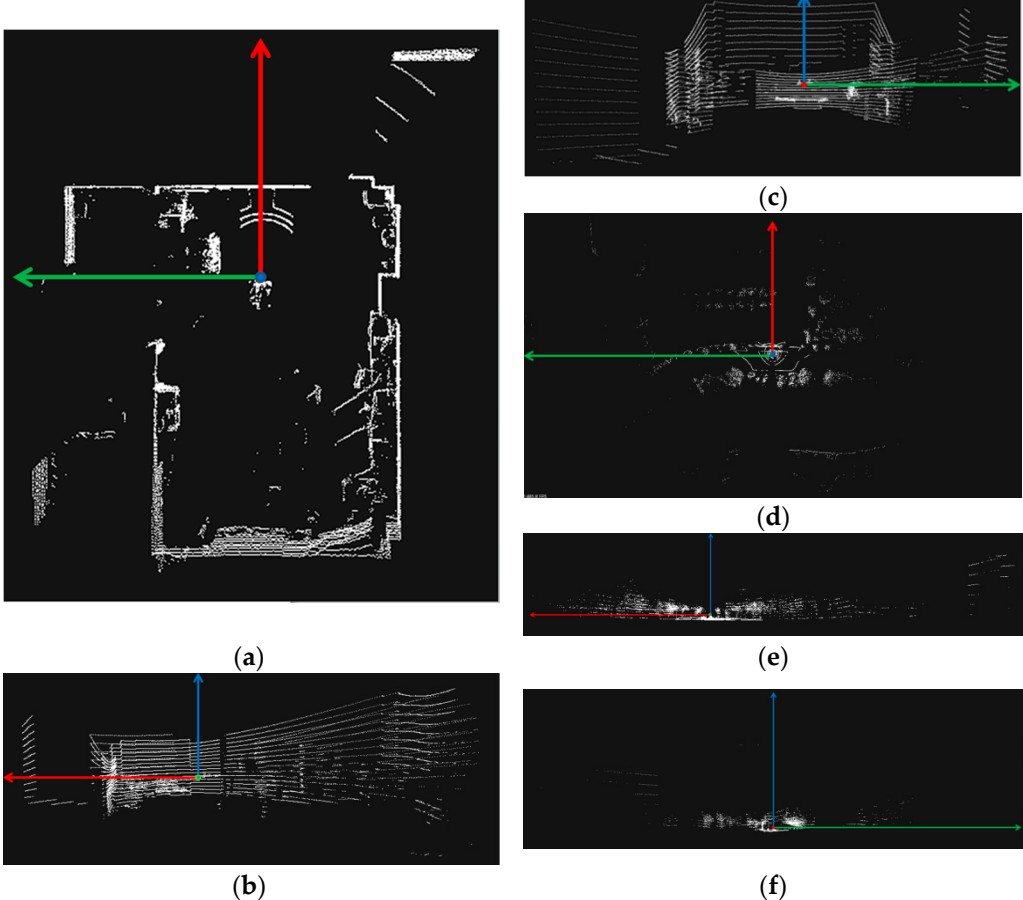

**Figure 2.** Raw point cloud image. (**a**) 40 cm × 40 cm *x-y* view; (**b**) 40 cm × 40 cm *y-z* view; (**c**) 40 cm × 40 cm *x-z* view; (**d**) 100 cm × 40 cm *x-y* view; (**e**) 100 cm × 40 cm *y-z* view; (**f**) 100 cm × 40 cm *x-z* view.

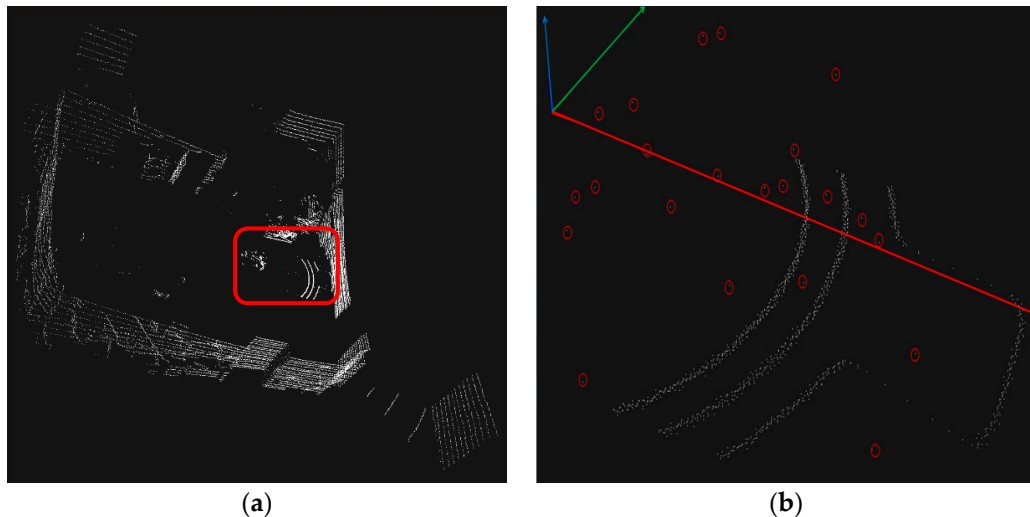

**Figure 3.** Before and after PassThrough filter comparison. (**a**) PassThrough filter the front point cloud data; (**b**) rendering after PassThrough filter.

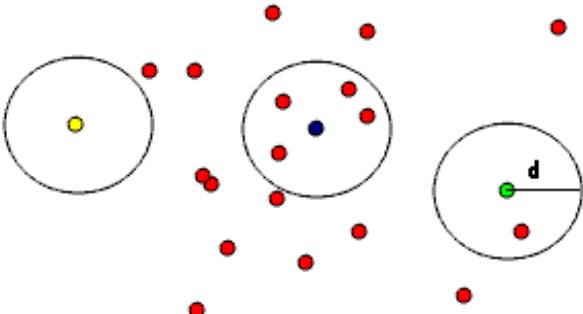

**Figure 4.** RadiusOutlier removal schematic.

We found that both the StatisticalOutlierRemoval filter and Conditional removal can achieve the results required in this experiment. However, compared to StatisticalOutlier-Removal, ConditionalRemoval is much more cumbersome, and RadiusOutlier removal is unsuitable for point clouds with negative obstacles. The point cloud is treated as a noise filter, but the real noise is not filtered out. The StatisticalOutlierRemoval filter is used to process point clouds in this paper.

The StatisticalOutlierRemoval filter is used to filter out points with non-conforming *z*-values and some outliers. For each point, we calculate the average distance from it to all its neighbors. It is assumed that the result is a Gaussian distribution, the result shape is determined by the mean and standard deviation, and that points whose mean distance is outside the standard range (defined by the mean and variance of the global distance) are defined as outliers and can be removed from the data set.

The negative obstacle point cloud after PassThrough filtering and the negative obstacle point cloud after StatisticalOutlierRemoval filtering in the PCL library are shown in Figures 3 and 5, respectively. The presence of significant outlier noise can be seen in Figure 3. After the noise is removed by the StatisticalOutlierRemoval filter (in Figure 5), the outlier noise is removed, and the negative obstacle point cloud pattern is well maintained.

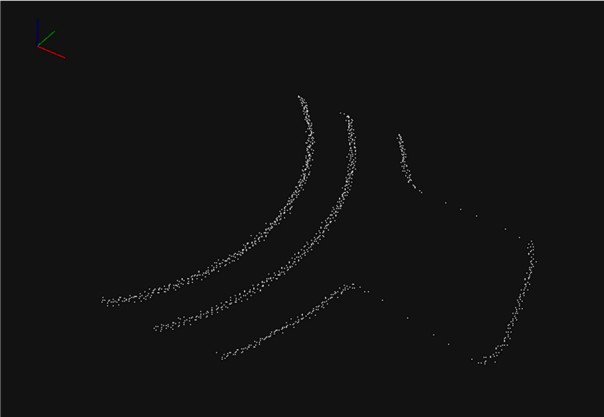

**Figure 5.** Rendering of negative obstacle point cloud after denoising.

### 2.3. Projection of 3D Point Cloud to 2D Image

The 3D point cloud collected by LiDAR is based on $(x, y, z)$ space coordinate point data calibrated in a rectangular space coordinate system containing an $x$, $y$, and $z$ We project the 3D point cloud along the $z$-axis onto the 2D plane formed by the $x$ and $y$. Considering that the point cloud captured by the Velodyne 16 LiDAR VLP-16 is the same as other ranges

scan data, points scanned by Velodyne can be roughly projected into a 2D image using the following projection function.

$$
\begin{aligned}
\theta &= \text{atan2}(y, x) \\
\phi &= \arcsin\left(z/\sqrt{x^2 + y^2 + z^2}\right) \\
r &= [\theta/\Delta\theta] \\
c &= [\phi/\Delta\phi]
\end{aligned}
\tag{1}
$$

where $p = (x, y, z)^T$ is the 3D point, $(r, c)$ is the position of its projected 2D image. $\theta$ and $\phi$ respectively represent the azimuth and elevation angle when observing the point. $\Delta\theta$ and $\Delta\phi$ represent the average horizontal and vertical angular resolution between the continuous beam emitters, respectively. The projection point map is similar to a histogram. We fill the elements at $(r, c)$ in the 2D point map with two-channel data $(d, z)$, where $d = \sqrt{x^2 + y^2}$. Note that $x$ and $y$ are coupled, and that $d$ denotes rotational invariance around $z$. The point of the projection is the same as the point of the observer. Few points are projected to the same 2D position, in which case the points closer to the observer are preserved. If no 3D points are projected at the 2D position, the element is populated with $(d, z) = (0, 0)$ [24–27].

### 2.4. Rasterize

To facilitate subsequent calibration studies and to take the maximum, minimum, or average values closest to the geometric features we estimate, we also rasterized the 2D image.

The most basic rasterization algorithm renders a 3D scene represented by a polygon onto a 2D surface. Fixed-scale raster processing is convenient and straightforward, but there is a contradiction between data volume and accuracy. A grid-scale that is too large decreases the accuracy of the detection, while a small scale increases the computational effort and defeats the purpose of gridding. The multi-scale grid can solve this problem well [28]. The rasterization process is shown in Figure 6.

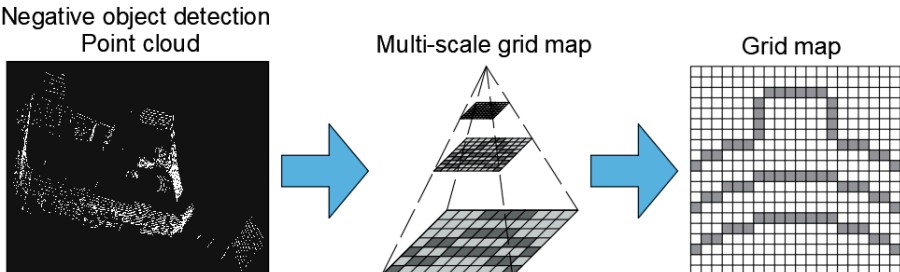

**Figure 6.** Rasterize.

Each small raster contains many three-dimensional pixel points, and the value of the largest point of the pixel point $z$ value in the raster with both length and width of 0.02 m is taken as the $z$ value of this raster to generate a single-channel two-dimensional planar image. In the point cloud data, the $z$-axis coordinates represent the depth of the point cloud, so the point cloud depth indicates the color's depth (the lighter the color, the larger the $z$-value, the darker the color, the smaller the $z$-value).

### 2.5. High Range Raster Image Generation

At this time, the $z$ value of the 3D point cloud was quantized as the RGB value in the range of 0~255 of the 2D raster image. 2.5D multi-scale grids are generated, counted each grid's characteristics, and marked the grid according to the characteristics. This generated an elevation raster image. The elevation raster image without grooves is shown in Figure 7a, and with grooves is shown in Figure 7b.

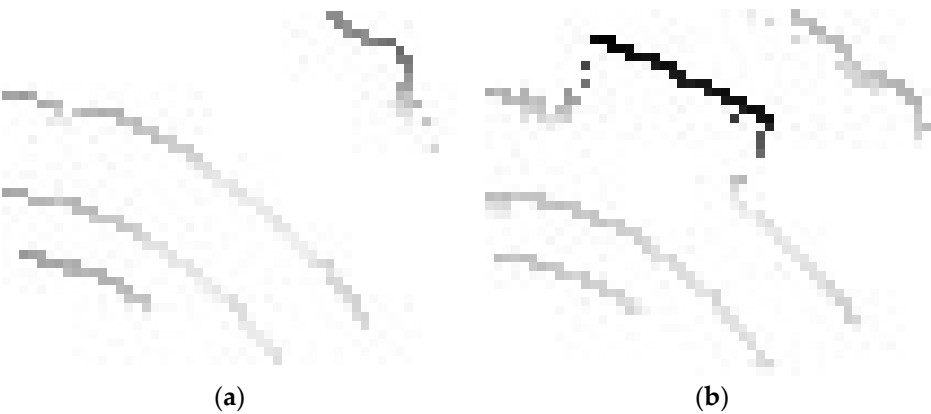

|     |     |
|-----|-----|
| (**a**) | (**b**) |

**Figure 7.** High range raster image. (**a**) Groove-less; (**b**) grooved.

*2.6. High Range Raster Image Denoising and Smoothing*

When generating an elevation raster image, it is inevitable that some new noise will appear on the image. Unlike the noise in Section 2.2, the noise in Section 2.2 is point cloud noise and needs to be processed by the above point cloud denoising filter. The noise of the elevation map is essentially different from the noise of the point cloud, so other methods are used to further denoise.

Noise burrs generally have definite frequency characteristics, and the use of appropriate filtering techniques can effectively suppress noise and improve the signal-to-noise ratio of the measurement system. The more commonly used smooth filtering methods are Median filter, Gauss filter, Aver filter, Adaptive filter, and Fit filter. The filter is to establish a mathematical model; through this model which shows the image data for energy conversion, low energy on the exclusion of noise is part of the low energy. For the eight-connected region of the image, the pixel value at the middle point is equal to the mean value of the pixel values in the eight-connected region, which will produce ringing in the image if the ideal filter is used. If Gauss filter is used, the system function is smooth and the ringing phenomenon is avoided.

In this paper, the three most commonly used filters for elevation raster images are selected for comparison: Median filter, Gauss filter, and Aver filter.

a.     Median filter.

The basic principle of the Median filter is to replace the value of a point in a digital image or sequence with the median value of each point in a neighborhood of that point, so that the surrounding pixels are close to the true value, thus eliminating isolated noise points. The method is used to generate a monotonically ascending (or descending) sequence of 2D data using a structured 2D sliding template that sorts the pixels within the plate by the pixel value. The output of the 2D Median filter is

$$g(x,y) = med\{(x - k, y - l), (k, l \in W)\} \tag{2}$$

where $f(x,y)$, $g(x,y)$, are the original image and the processed image, respectively. $W$ is a 2D template, usually a $2 \times 2$, $3 \times 3$ area, but can also be different shapes, such as a line, circle, cross, ring, rhombus and so on.

b.     Gauss filter.

A Gauss filter is essentially a kind of signal filter; its use is the signal smoothing process is to achieve a better image edge. The image is first Gauss-smoothed filtered as well as noise removed, and then the second-order derivative is found to determine the edge, which is also calculated as a frequency-domain product to a null-domain convolution. Set as a given point, its neighborhood is $\{P_{i,j} = (x_{i,j}, y_{i,j}, z_{i,j}) | -n \leq i \leq n, -m \leq j \leq m\}$,

where $x_{i,j} = i\Delta x$, $y_{i,j} = j\Delta y$, $z_{i,j} = f(x_{i,j}, y_{i,j})$. After Gauss filter smoothing the $P_{0,0}$ point after the $z$-axis directional coordinates of the $P_{0,0}$ point is

$$\overline{z}_{0,0} = \frac{1}{c}\sum_{i=-n}^{n}\sum_{j=-m}^{m} z_{-i,-j}g_{i,j}\Delta x\Delta y \tag{3}$$

where $g(i, j)$ is the Gaussian function, and $c$ is the normalization coefficient, which are expressed as

$$\begin{aligned} g_{i,j} &= \frac{1}{2\pi\sigma^2}e^{-(i^2\Delta x^2 + j^2\Delta y^2)/2\sigma^2}\\ c &= \sum_{i=-n}^{n}\sum_{j=-m}^{m} g_{i,j}\Delta x\Delta y \end{aligned} \tag{4}$$

c.    Aver filter.

The idea of the Aver filter is to replace the value of a given point with a weighted average value within the neighborhood of the given point. Set $P_{0,0}$ for a given point, its neighborhood is $\{P_{i,j} = (x_{i,j}, y_{i,j}, z_{i,j}) | -n \le i \le n, -m \le j \le m\}$. After the Aver filter smoothing point $P_{0,0}$ becomes

$$\overline{P}_{0,0} = \sum_{i=n}^{n}\sum_{j=-m}^{m} h_{i,j}p_{i,j} \tag{5}$$

where $h_{i,j}$ is the normalized weighting factor.

After comparing the three filtering denoising principles and methods, we found that the Median filter method is very useful in eliminating particle noise. It has a unique role in the phase analysis and processing method of optical measurement fringe images, but it has little effect on the fringe center analysis method. The Median filter is a classical method for smoothing out the noise and is commonly used to protect edge information in image processing. To further determine whether the filter we chose is appropriate for this experiment, we de-noise the elevation raster images using three separate filters. Figure 8 shows the comparison of negative obstacles before denoising and the three kinds of filtering effects. From left to right, they are the image before denoising, the denoising effect of the Aver filter, the denoising effect of the Gauss filter, and the denoising effect of the Median filter. It can be seen that the denoising effect of the Aver filter is not significant, and the Gauss filter redundantly removes the pixels of the negative obstacle, which will affect the subsequent estimation of the length of the negative obstacle. Only the denoising effect of the Median filter is most suitable for this method. Therefore, the OpenCV-based Median filter algorithm was finally chosen to denoise our elevation raster images.

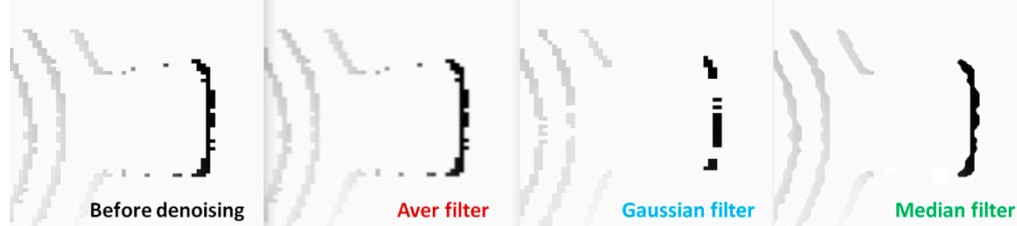

**Figure 8.** Three types of filtering and denoising effect comparison chart.

Based on the premise of the Median filter, this paper adopts the design idea, necessary operation steps, algorithm flow, and algorithm analysis of a fast Median filter encoding algorithm, which takes advantage of the position relationship of the elements in the data window and considers the correlation of the data elements in two adjacent Median filter windows. It uses the encoding sorting method to retain the encoding sorting information of the previous window data as a reference for the data sorting in the following window. This

algorithm reduces the number of comparisons in the Median filter process by combining two adjacent Median filter operations in a traditional algorithm into one.

## 3. Estimation of Geometric Features of Structured Negative Obstacles

ALV/UGV research is usually divided into the structured pavement, semi-structured pavement, and unstructured pavement. Structured pavement refers to a horizontal road environment with characteristics of typical road edge rules, chromatic aberration characteristics, and street line characteristics, such as highways, national highways, provincial highways, and urban roads. In a structured environment, the road surface of an ROS AUTONOMOUS NAVIGATION VEHICLE is relatively horizontal, VLP-16 LiDAR, the measurement conversion based on the vehicle body coordinate plane, and the *z*-axis threshold control method can effectively detect negative obstacles. However, since our goal is to estimate a structured negative obstacle's geometry, we need to go a step further and accurately estimate a negative obstacle's geometry based on algorithms such as the Progressive Probabilistic Hough Transform and geometric methods mathematical models.

### 3.1. Length Estimation

The length estimation method adopts the Probabilistic Hough Transform method [29]. The standard Hough Transform essentially maps the image to its parameter space. It needs to calculate all M edge points so that its computation and memory space will be much larger. As such, we mapped the 3D point cloud image to the 2D image and determined the *z*-value of the color (the lighter the color, the larger the *z*-value; the darker the color, the smaller the *z*-value). If only m (m < M) edge points are processed in the input image, then selecting these m edge points is probabilistic, so the method is called Probabilistic Hough Transform. Another essential feature of this method is that it can detect the end of the line; that is, it can detect the straight line's two endpoints in the image and accurately locate its straight line. The HoughLinesP function was used to utilize Progressive Probabilistic Hough Transform to detect straight lines. The specific steps are:

(1)   Randomly select one feature in the elevation raster image, i.e., the edge point where the laser hits the back wall of the negative obstacle, and if that point has been identified as a point on a straight-line, continue to randomly select an edge point from the remaining edge points until all edge points have been selected;

(2)   Perform a Hough Transform [30] on the point and perform a cumulative calculation;

(3)   Select the point with the most considerable value in the Hoff space, and if the point is larger than the threshold, proceed to step 4, and if the point is smaller than or equal to the threshold, return to step 1;

(4)   According to the maximum value obtained from the Hough Transform, from that point, the line is displaced in the direction of the line to find the two endpoints of the line;

(5)   Calculate the length of the straight-line. If it is greater than a certain threshold, it is just regarded as a straight-line output and goes back to step 1.

The above are the steps we use Progressive Probabilistic Hough Transform to measure the length of the negative obstacle b in the 2D raster image. The flowchart is shown in Figure 9.

As shown in Figure 10, the negative obstacle's elevation raster image contains b pixels (grid), which is the distance between the two points *p* and *q*. Since each pixel of the elevation raster image represents a raster with a length and a width of 0.02 m, we can estimate the actual length B of the negative obstacle to be 0.02b based on the scale.

### 3.2. Mathematical Model for Width Estimation

As shown in Figure 11, we estimate the width of the negative obstacle through the median value of Figure 8. After filtering and denoising the image, it is known *p* and *q* two point coordinates are $(x_1, y_1)$, $(x_2, y_2)$, over the negative obstacle height raster image of

the two edge points $p$ and $q$ to make a straight line for the line $a$, from which we can get a straight line $a$ equation for the

$$(x - x_1)(y_2 - y_1) = (y - y_1)(x_2 - x_1)$$
$$y = \frac{y_2 - y_1}{x_2 - x_1} x - \frac{x_1 y_2 - x_2 y_1}{x_2 - x_1}$$

(6)

where the slope $k_a$ of the line $a$ is $\frac{y_2 - y_1}{x_2 - x_1}$.

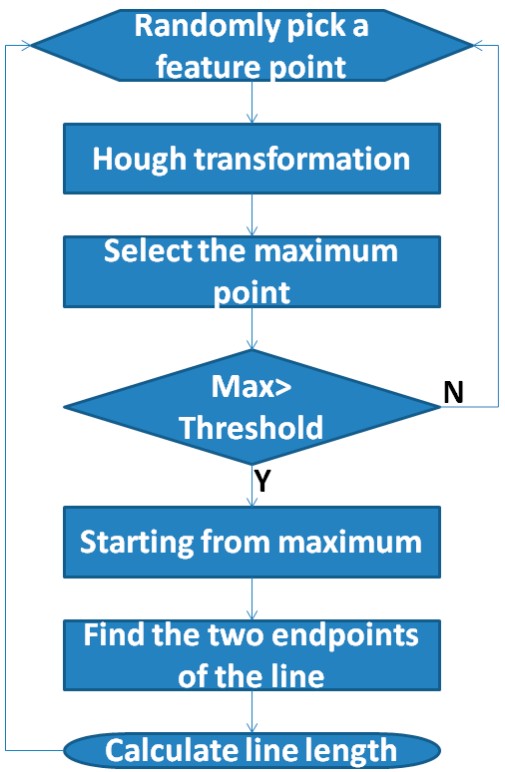

**Figure 9.** Progressive Probabilistic Hough Transform detects straight-line data.

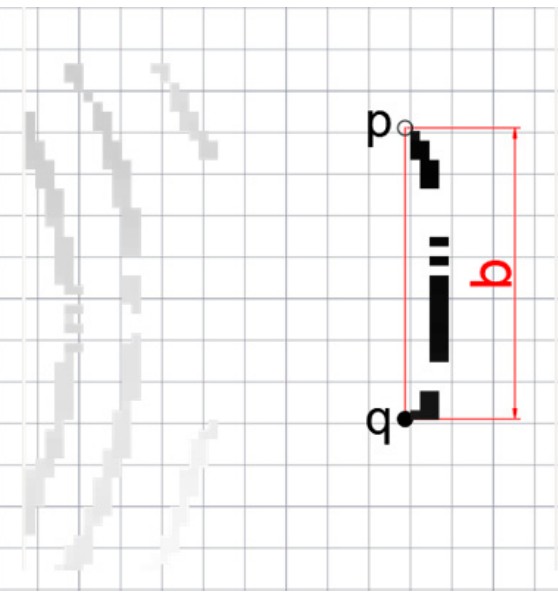

**Figure 10.** Estimating the length of negative obstacles.

Make the perpendiculars of the line a through points $p$ and $q$ respectively, so that the normal through point $p$ is $b1$ and the normal through point $q$ is $b2$; by the above information the equations of the lines $b1$ and $b2$ are

$$b1 : y = \frac{x_2-x_1}{y_1-y_2}x - \frac{x_1x_2-x_1^2}{y_1-y_2} + y_1$$
$$b2 : y = \frac{x_2-x_1}{y_1-y_2}x - \frac{x_2^2-x_1x_2}{y_1-y_2} + y_2$$

(7)

where the slope $k_{b1}$, $k_{b2}$ of the line $b1$, $b2$ is $\frac{x_2-x_1}{y_1-y_2}$.

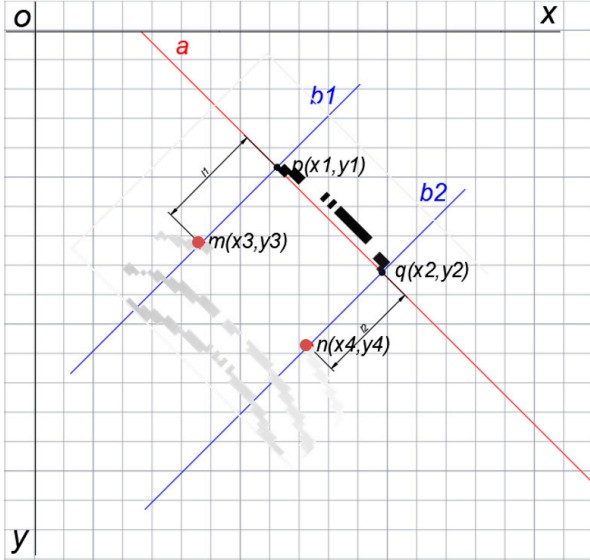

**Figure 11.** The mathematical model for width estimation.

Taking two points $p$ and $q$ as starting points, OpenCV is used to carry out pixel traversal on the two lines $b1$ and $b2$. When a pixel point that meets the requirements (pixel value less than 255) is retrieved, the traversal is stopped immediately and the coordinates of this point are recorded. The point on line of $b1$ is set as $m$ and the point on line $b2$ are set at $n$. Determine the coordinates of $m$ and $n$ as $(x3, y3)$ and $(x4, y4)$, as shown in Figure 11. Connect the two line segments $pm$ and $qn$, and calculate the lengths of $pm$ and $qn$ as $l1$ and $l2$ respectively. $pm$ and $qn$ are the width of negative obstacles in the elevation raster image.

To make the experimental results more accurate, we take the average of $l1$, $l2$ and make it $l$.

$$L = \frac{l1 + l2}{2}$$

(8)

$L$ is the result of the pixel width of the negative obstacle that we finally estimate on the elevation raster image. Since each pixel of the elevation raster image represents a raster with a length and a width of 0.02 m, we can estimate the actual width $L$ of the negative obstacle to be $0.02l$ according to the scale.

## 4. Experiments

This experimental code is based on data provided by the Velodyne VLP-16 LiDAR presented above, and written in RobWare Studio. The working platform uses a computer with ubuntu 16.04 LTS, Intel i5 Processor, 2.30 GHz, and 8 GB of RAM.

### 4.1. Experimental Platform and Test Environment

To accurately estimate the geometric characteristics of structured negative obstacles, this paper uses the laboratory ROS Autonomous Navigation Vehicle equipped with Velodyne VLP-16 LIDAR as the experimental platform for experiments. The ROS Autonomous

Navigation Vehicle is shown in Figure 12. The explicit parameters of the experiment are shown in Table 1.

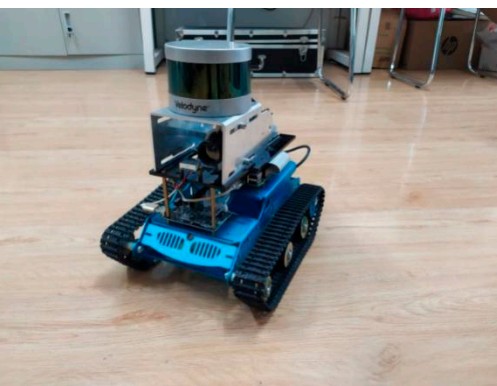

**Figure 12.** ROS Autonomous Navigation Vehicle.

**Table 1.** Specific parameters of the experiment.

| Parameter | Technical Indicators |
|---|---|
| LiDAR Height (m) | 0.3 |
| Effective Scanning Angle (°) | −15~+15 |
| Vehicle Speed (m/s) | 1 |
| Sampling Frequency (Hz) | 20 |

Multiple sets of data were collected in a structured environment for estimating the geometric features of negative obstacles, and the performance metrics for evaluating this algorithm are mainly the measurements of the length and width of negative obstacles. A negative obstacle's size and shape affect the detection's accuracy, with more extensive geometric features of negative obstacles being estimated with higher accuracy and vice versa. The length of the negative obstacle is defined as B (the width of the negative obstacle in the trolley direction), and the width is L (the width of the negative obstacle in the vertical section of the trolley). The main parameters and specifications of the VLP-16 LiDAR are shown in Table 2.

**Table 2.** LiDAR main parameters and technical specifications.

| Parameter | Technical Indicators |
|---|---|
| Number of Laser Lines | 16 |
| Measuring range (m) | 100 |
| Weight (g) | 830 |
| Measurement Accuracy (cm) | ±3 |
| Horizontal Measurement Angle Range (°) | 360 |
| Horizontal Angle Resolution (°) | 0.1~0.4 |
| Vertical Angle Resolution (°) | 2 |
| Vertical Measurement Angle Range (°) | 30 (−15~+15) |
| Detection Frequency (Hz) | 5~20 |

According to the point cloud distribution density of the radar at different distances and the size of the target negative obstacle, the grid map is divided into two different sizes, 2 cm × 2 cm and 5 cm × 5 cm, to estimate the geometric characteristics of the negative obstacle as accurately as possible, While reducing the amount of calculations, this paper uses a 5 cm × 5 cm grid before the point cloud denoising, and the elevation grid image uses a 2 cm × 2 cm grid after denoising and smoothing. A typical negative obstacle scenario for the data collected in this experiment is illustrated in Figure 13. The unstructured experimental scenario is shown in Figure 14, where the maximum diameter of Figure 14a is 1 m; the maximum diameter of Figure 14b is 1.5 m; and the maximum

diameter of Figure 14c can cover the whole LiDAR detection range, so we mainly estimate its width.

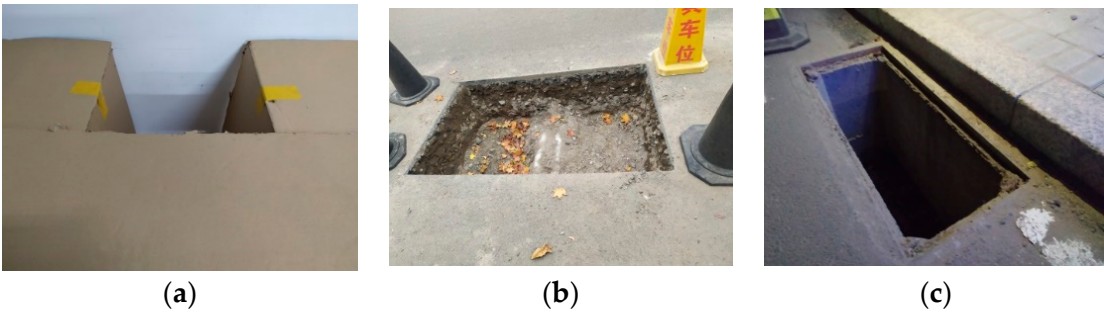

| (a) | (b) | (c) |

**Figure 13.** Typical negative obstacle scenario. (**a**) 40 cm × 40 cm; (**b**) 100 cm × 40 cm; (**c**) 120 cm × 60 cm.

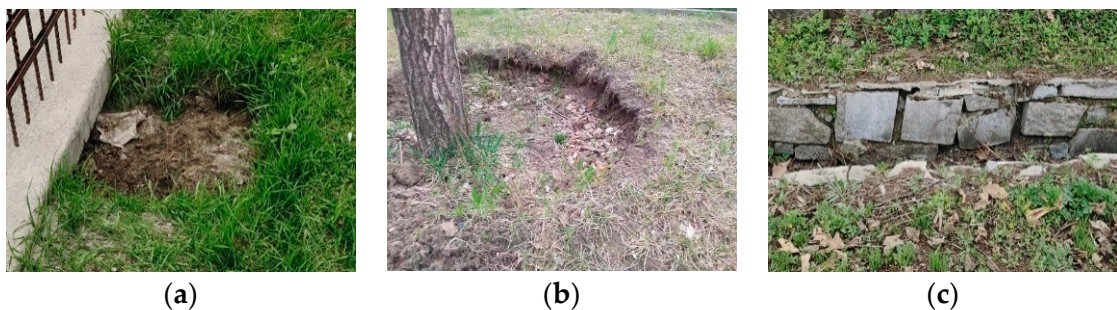

| (a) | (b) | (c) |

**Figure 14.** Unstructured experimental scenarios. (**a**) 1 m; (**b**) 1.5 m; (**c**) ∞.

*4.2. Validation Experiment*

4.2.1. Structured Environment

This paper verifies the experimental measurement's accuracy by comparing the manual marking's geometric features with the experimental measurement's geometric features and verifying the negative obstacle estimation accuracy in many cases. First, this experiment analyzed and verified the size of the negative obstacle on geometric feature estimation accuracy. Three types of negative obstacles were selected, as shown in Figure 13. Figure 13a is a 40 cm × 40 cm square negative obstacle. Figure 13b is a 100 cm × 40 cm rectangular negative obstacle, and Figure 13c is a 120 cm × 60 cm negative obstacle. Secondly, we wished to verify the distance between the LiDAR and the negative obstacle to estimate accuracy. The detection range of the Velodyne VLP-16 LiDAR was 100 m, but the detection range of negative obstacles was far from 100 m, so we needed to calculate the detection range of negative obstacles ourselves based on the detection principle of the LiDAR.

According to known parameters, the LiDAR was mounted on the ROS Autonomous Navigation Vehicle with a height of 0.3 m. The vertical measurement angle range of the VLP-16 LiDAR was −15°~+15°, and there was a laser scanning beam every 2°. Figure 15 shows the establishment of a distance measurement method schematic diagram based on known parameters.

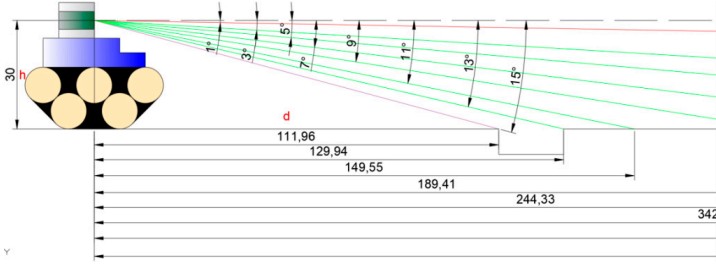

**Figure 15.** Schematic diagram of negative obstacle detection range estimation.

LiDAR $-15°$ laser beam (purple line) was the nearest detection distance, and $-1°$ laser beam (red line) was the farthest scanning distance; thus, we could estimate the negative obstacle detection range of LiDAR from 1.1~17.2 m under stationary conditions.

$$\begin{pmatrix} d_1 \\ d_2 \\ d_3 \\ d_4 \\ d_5 \\ d_6 \\ d_7 \\ d_8 \end{pmatrix} = h \times \begin{pmatrix} \cot\theta_1 \\ \cot\theta_2 \\ \cot\theta_3 \\ \cot\theta_4 \\ \cot\theta_5 \\ \cot\theta_6 \\ \cot\theta_7 \\ \cot\theta_8 \end{pmatrix} \tag{9}$$

where $d_1 \sim d_8$ is the distance between the downward eight lines launched to the ground and the cart in the distance measurement method of negative obstacle detection range estimation, input $h$ to the model as 30, input $(\theta_1\ \theta_2\ \theta_3\ \theta_4\ \theta_5\ \theta_6\ \theta_7\ \theta_8)$ as $(1\ 2\ 3\ 4\ 5\ 6\ 7\ 8)$ respectively finally get $d_1 \sim d_8$ as 111.96 cm, 129.94 cm, 149.55 cm, 189.41 cm, 244.33 cm, 324.9 cm, 497.32 cm, 1718.7 cm.

According to the calculated range, this experiment was equipped with LiDAR on an ROS Autonomous Navigation Vehicle, starting measurements from 20 m away from the three negative obstacles at a speed of 1 m/s and a measuring frequency of 20 Hz, and recording the experimental data. The LiDAR recorded 30 frames of data per second, and it took 20 s for a 20 m distance to record 600 frames of data. Three sets of experiments in total recorded 1800 frames of data. Each group of experiments selected each frame of data in which the laser scanning beam hit the edge of the front wall of the negative obstacle from 600 frames of data. Each group of experiments collected eight groups of valid data.

Figure 16 shows the results of detecting a negative obstacle to a size of $100 \times 40$ cm at different distances, based on the distance measurement method of Figure 15. Figure 16a–h are respectively 111.96 cm, 129.94 cm, 149.55 cm, 189.41 cm, 244.33 cm, 324.9 cm, 497.32 cm, and 1718.7 cm structured pavement field scenes (left side), and LiDAR detection point cloud.

Each set of data measures the length and width of negative obstacles, and the statistical results are shown in Table 3.

To verify the accuracy of the method in this paper, an error analysis was performed on the data collected in Table 3. The absolute error was calculated from the data in Table 3, which was the measured value minus the actual value. In order to verify the relationship between the error and the size of the negative obstacles, and because the length and width of each negative obstacles were different, it is not enough for us to calculate the absolute error alone, so we further calculated the relative error based on the absolute error, i.e., the absolute error as a percentage of the actual negative obstacles size, so that we could have a more intuitive understanding of the relationship between the size of the negative obstacles and the error. As showed in Table 4, where ALE is the absolute length error, RLE is the relative length error, AWE is the absolute width error, and RWE is the relative width error. This experiment compared the size relationship between the relative errors of length 40 cm, 100 cm and 120 cm and the size relationship between the relative errors of width 40 cm and 60 cm at a certain distance from the negative obstacles, respectively, and the trend of the relative errors at a certain length and width of the negative obstacles is shown in Figure 17, where Figure 17a is the length relative error analysis, and Figure 17b is the width relative error analysis. Figure 18 shows the trend of the absolute error in the case of different lengths of negative obstacles and different distances from the negative obstacles, with error bars to help the reader to understand the error relationship more clearly and intuitively.

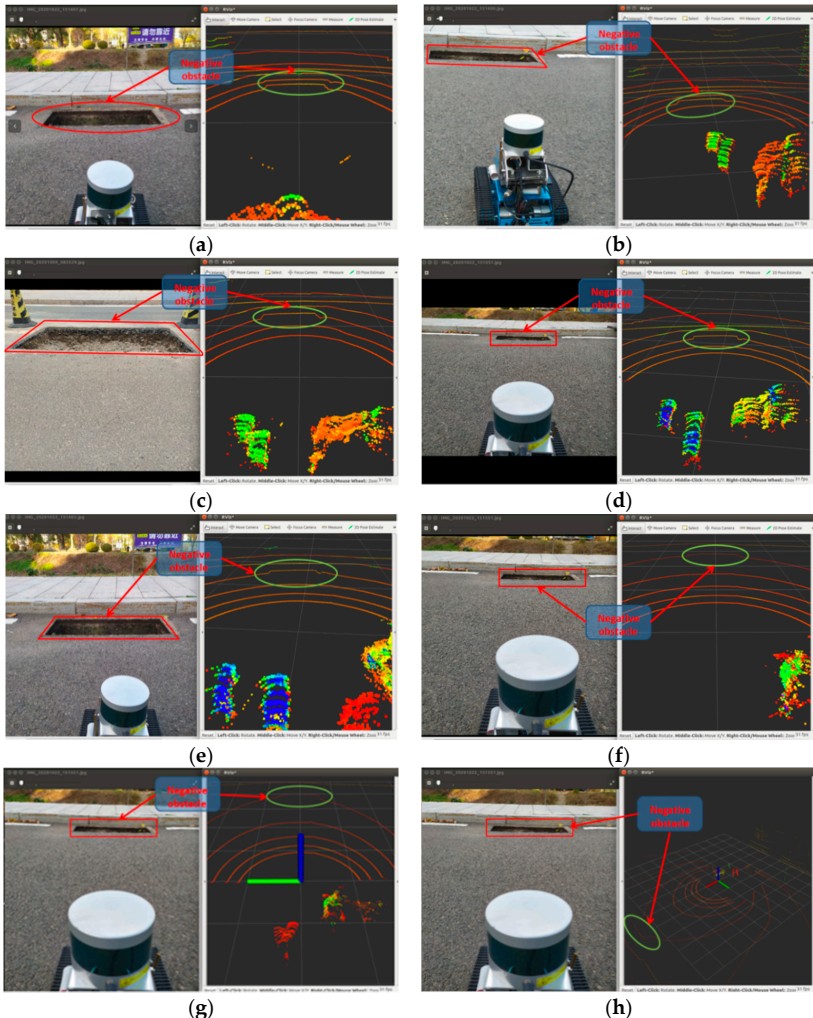

**Figure 16.** Comparison of detection results at different distances. (**a**) 111.96 cm; (**b**)129.94 cm; (**c**) 149.55 cm; (**d**)189.41 cm; (**e**) 244.33 cm; (**f**) 324.90 cm; (**g**) 497.32 cm; (**h**) 1718.7 cm.

**Table 3.** Experimental results of multidimensional, multi-group negative obstacle measurements.

| Experiment No. | Distance (cm) | Hand Mark | | Experimental Measurement | |
|---|---|---|---|---|---|
| | | Length (m) | Width (m) | Length (m) | Width (m) |
| 1 | <111 | 0.4 | 0.4 | - | - |
| | 111.96 | | | 0.4 | 0.4 |
| | 129.94 | | | 0.400143 | 0.400567 |
| | 149.55 | | | 0.4032 | 0.40416 |
| | 189.41 | | | 0.4087 | 0.40772 |
| | 244.33 | | | 0.4123 | 0.41497 |
| | 342.90 | | | 0.4225 | 0.4357 |
| | 497.32 | | | 0.36491 | 0.370384 |
| | 1718.70 | | | 0.32 | 0.3004 |
| | >1719 | | | - | - |
| 2 | <111 | 1 | 0.4 | - | - |
| | 111.96 | | | 1 | 0.4 |
| | 129.94 | | | 1 | 0.4 |
| | 149.55 | | | 1.000212 | 0.40034 |
| | 189.41 | | | 1.0008 | 0.40087 |
| | 244.33 | | | 1.00443 | 0.4047 |
| | 342.90 | | | 1.00753 | 0.4192 |
| | 497.32 | | | 1.01847 | 0.4345 |
| | 1718.70 | | | 1.0456 | 0.47432 |
| | >1719 | | | - | - |

**Table 3.** *Cont.*

| Experiment No. | Distance (cm) | Hand Mark | | Experimental Measurement | |
|---|---|---|---|---|---|
| | | Length (m) | Width (m) | Length (m) | Width (m) |
| 3 | <111 | 1.2 | 0.6 | - | - |
| | 111.96 | | | 1.2 | 0.6 |
| | 129.94 | | | 1.2 | 0.6 |
| | 149.55 | | | 1.200333 | 0.601333 |
| | 189.41 | | | 1.2004 | 0.60384 |
| | 244.33 | | | 1.201332 | 0.59357 |
| | 342.90 | | | 1.2078 | 0.62 |
| | 497.32 | | | 1.210322 | 0.620384 |
| | 1718.70 | | | 1.24 | 0.540357 |
| | >1719 | | | - | - |

**Table 4.** Length and width error analysis table.

| Distance (cm) | | 111.96 | 129.94 | 149.55 | 189.41 | 244.33 | 342.9 | 497.32 |
|---|---|---|---|---|---|---|---|---|
| 40 cm | ALE | 0 | 0.0143 | 0.32 | 0.87 | 1.23 | 2.25 | −3.509 |
| | RLE | 0% | 0.03575% | 0.8% | 2.175% | 3.075% | 5.625% | −8.7725% |
| 100 cm | ALE | 0 | 0 | 0.0212 | 0.08 | 0.443 | 0.753 | 1.847 |
| | RLE | 0% | 0% | 0.0212% | 0.08% | 0.443% | 0.753% | 1.847% |
| 120 cm | ALE | 0 | 0 | 0.0333 | 0.04 | 0.1332 | 0.78 | 1.0322 |
| | RLE | 0% | 0% | 0.02775% | 0.033% | 0.111% | 0.65% | 0.86017% |
| 40 cm | AWE | 0 | 0.0567 | 0.416 | 0.772 | 1.497 | 3.57 | −2.9616 |
| | RWE | 0% | 0.14175% | 1.04% | 1.93% | 3.7425% | 8.925% | −7.404% |
| 60 cm | AWE | 0 | 0 | 0.1333 | 0.384 | −0.0634 | 2 | 2.0384 |
| | RWE | 0% | 0% | 0.22217% | 0.64% | 0.10567% | 3.33% | 3.3973% |

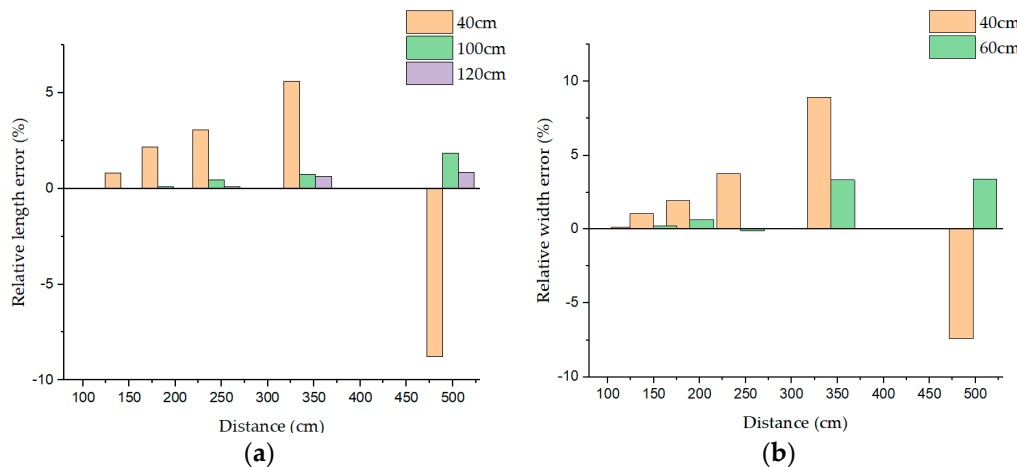

**Figure 17.** Relative error analysis. (**a**) Length relative error analysis; (**b**) width relative error analysis.

According to the data of Tables 3 and 4 and Figures 17 and 18, we can see that in the case of a certain distance from the negative obstacles, the longer the length or the wider the width of the negative obstacles, the smaller the estimated geometric feature error, and the shorter the length or the narrower the width, the larger the estimated geometric feature error; in the case of a certain length and width of the negative obstacle, the further the distance from the negative obstacles, the larger the estimated geometric feature error, and the closer the error. The closer the distance, the smaller the error. From this we conclude that the detection effect of this experiment on the geometric features of negative obstacles is positively correlated with their length and width, i.e., the longer the length of negative obstacle (the wider the width) the higher the detection accuracy and the smaller the error, and vice versa: the lower the detection accuracy the larger the error. The accuracy of this method in detecting the geometric features of negative obstacles is negatively correlated

with the distance between the LiDAR and the negative obstacles, i.e., the farther the LIDAR is from the negative obstacles, the lower the detection accuracy and the larger the error, and vice versa: the higher the detection accuracy and the smaller the error.

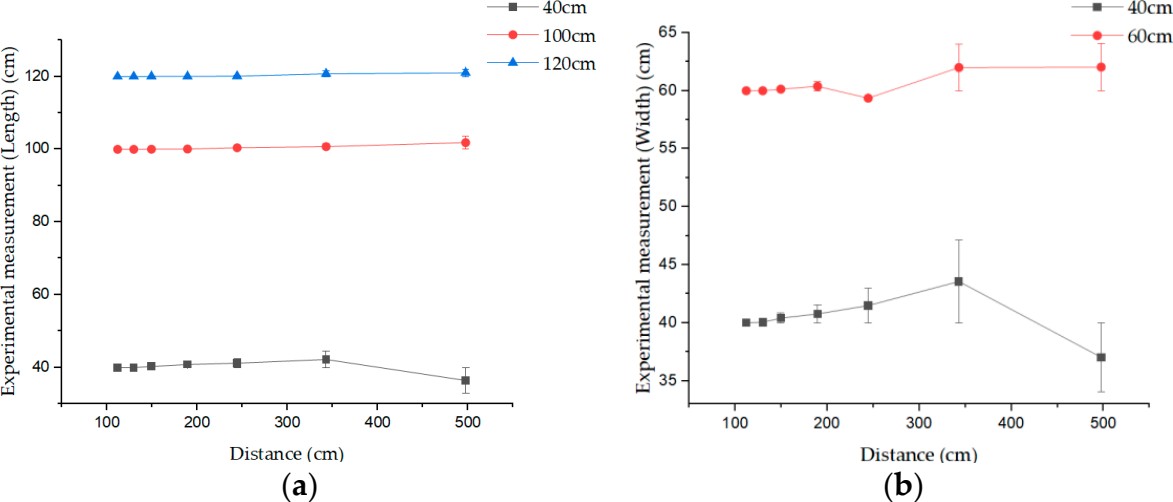

**Figure 18.** Verify the data error bar. (**a**) Verify the length data error bar; (**b**) verify the width data error bar.

4.2.2. Unstructured Environment

Further, to verify the applicability of the present method in different environments, this experiment also detected negative obstacles in an unstructured environment and tried to estimate their geometric characteristics. Since the negative obstacle of the non-institutionalized pavement were irregular, it was difficult to define their lengths and widths. According to the applicability of this method, the maximum diameter of the negative obstacle is taken as the criterion for measurement and estimation. The results of the detection of the non-structural environment are shown in Table 5.

**Table 5.** Experimental results of negative obstacle measurement in non-structural environments.

| Experiment No. | Distance (m) | Maximum Diameter Measurable (m) | Measurement Results (m) | Error (%) |
|---|---|---|---|---|
| | 1 | | - | - |
| | 2 | | 0.8479 | 28.75 |
| | 3 | | 0.9432 | 20.74 |
| 1 | 4 | 1.19 | 0.5014 | 57.87 |
| | 5 | | 0.6470 | 45.63 |
| | 6 | | - | - |
| | 1 | | - | - |
| | 2 | | 1.9502 | 20.38 |
| | 3 | | 2.2335 | 37.87 |
| 2 | 4 | 1.62 | 0.8274 | 48.97 |
| | 5 | | 1.1193 | 30.91 |
| | 6 | | - | - |
| | 1 | | - | - |
| | 2 | | 0.4735 | 5.3 |
| | 3 | | 0.5596 | 11.92 |
| 3 | 4 | 0.5 | 0.4324 | 13.52 |
| | 5 | | 0.3350 | 33 |
| | 6 | | - | - |

According to the data in Table 5, we can see that the detection range and accuracy vary in different non-structural environments, but in the case of no obstruction in front of the negative obstacle, the negative obstacle within 5 m from the LiDAR can be detected. Due to the uneven terrain in the nonstructural environment, the geometric characteristics of the negative obstacles have large errors and no regularity.

*4.3. Comparison Experiment*

To further verify the accuracy of the proposed method in this paper. It is compared with two typical negative obstacle detection methods. One of the papers [9] is a negative obstacle detection algorithm based on the fusion of single-line LiDAR and monocular vision. The literature [2] is a negative obstacle detection method based on infrared features. As well as this, the literature [3] further couples thermal feature-based cues and geometric retrieval based on stereo distance data for nighttime negative obstacle detection. At the same time, to verify the accuracy of the estimation method of the negative obstacle geometric features proposed in this paper, the data estimated based on the mathematical model in this paper are also counted, and subjected to error analysis. For comparison purposes, only the errors within the detection limits of each method are counted. In order to objectively compare the relationship between the measurement errors of each method, we took the middle value for all experimental environments, i.e., the statistics were conducted under the condition that the length of the negative obstacle was 100 cm and the width was 40 cm. Due to the high testing conditions in the literature [2,3], the comparison experiments were performed in two groups: daytime and nighttime.

From Figure 19, it can be found that the negative obstacle detection performance of the method in this paper is better than the methods in the literature [2,3,9], both in daytime and nighttime. We can see that the detection error of the VLP-16 LiDAR-based method in this paper does not change much in the two sets of experiments at day and night, while the error of the method in the literature [9] at night is significantly larger than that at daytime, which is because the detection capability of the monocular sensor at night will be greatly discounted, leading to the fact that the method in the literature [9] at night is almost equal to the detection of negative obstacles by single-line LiDAR only, and the single-line. The real-time and stability of LiDAR is much lower than that of 16-line LiDAR. Figure 20a for single-line LiDAR to detect negative obstacles, (b) for 16-line LiDAR to detect negative obstacles) shows the gap between the point cloud map generated by single-line LiDAR and multi-line LiDAR to detect negative obstacles under the condition of the same environment and the same distance, obviously multi-line LiDAR generated. The point cloud is more easily recognized by the machine. The method in the literature [2,3] is based on the phenomenon that negative obstacles tend to dissipate less heat and higher temperatures than the surrounding terrain at night, and local intensity analysis is performed on the IR images to mark significant intensity regions as potential negative obstacle regions, and then the final negative obstacle confirmation is performed by multi-frame verification and fusion. The biggest limitation of this method is that it can only detect negative obstacles at night, and is affected by the weather environment relatively significantly, while there will also be animals and other living things mistakenly detected as negative obstacles, and the detection error at night is also greater than the error of this method.

The experimental results show that when the distance between LiDAR and negative obstacles is less than 111 cm or more than 1719 cm, LiDAR cannot detect negative obstacles, which belong to the detection blind area of this method, and the error of detecting negative obstacles in unstructured pavement is greater compared to structured pavement because the sensor is more single. The method of Yunfei Cai [31] will be referred to in the subsequent study. At the same time, the speed of the trolley cannot be too fast; when the speed of the trolley is greater than 1 m/s, in 20 Hz sampling rate, even a flat road will miss detection of smaller negative obstacles (diameter less than 15 cm), while reducing the detection accuracy of larger negative obstacles (diameter greater than 15 cm).

Overall, the present method is very accurate in estimating the geometric features within 5 m from the negative obstacles, and the driving speed of the unmanned vehicle (1 m/s). A driving time of 5 m is fully capable of timely obstacle avoidance. Therefore, the practicality of this method is very strong.

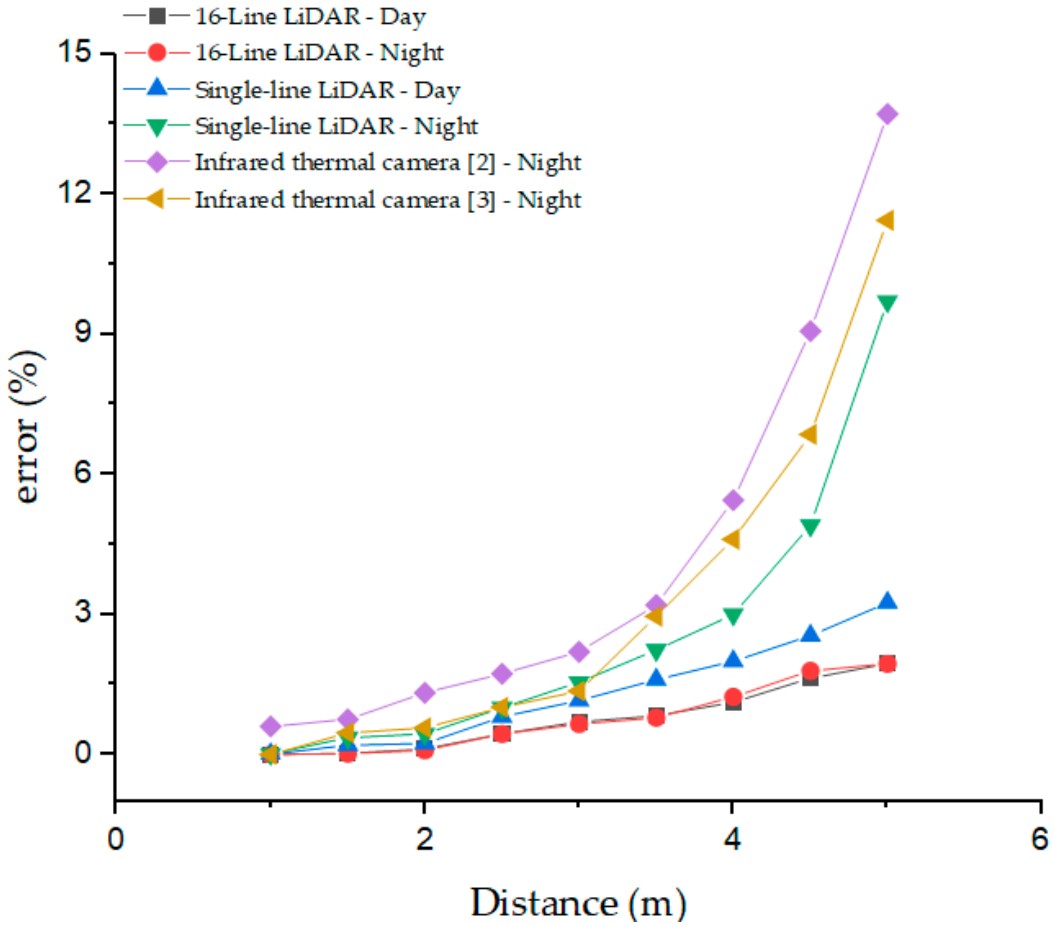

**Figure 19.** Comparison diagram of negative obstacle detection performance.

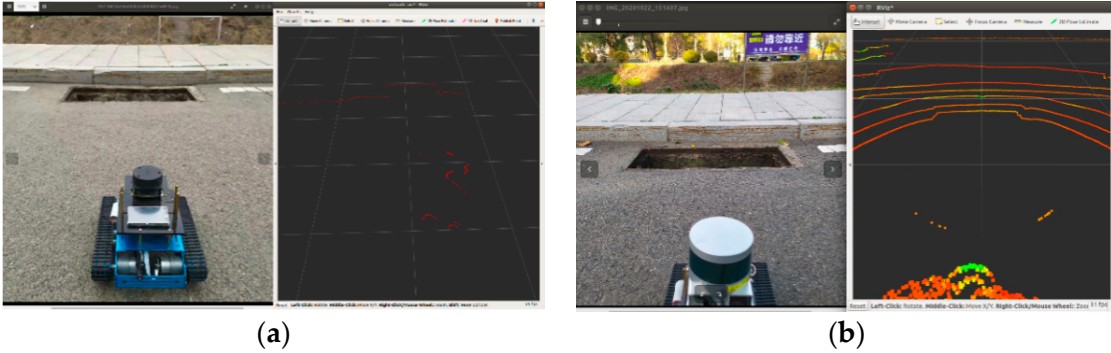

|(**a**)|(**b**)|

**Figure 20.** Performance comparison of single-line LiDAR and 16-line LiDAR. (**a**) Single line LiDAR detects negative obstacle; (**b**) 16-line LiDAR detects negative obstacle.

## 5. Conclusions

This paper proposes estimating geometric features of negative obstacles based on a single-frame 3D laser point cloud. The method is composed of two parts: negative obstacle location and geometric feature estimation. The point cloud denoising (PassThrough filter to locate the precise position of the negative obstacle point cloud, StatisticalOutlierRemova filter to handle the noise, Median filter to smooth the denoising) presents an accurate 2D negative obstacle image, improving the intuitiveness and accuracy of the detection. The estimation method is based on a mathematical model combined with Progressive Probabilistic Hough Transform and OpenCV pixel traversal, doing multi-feature correlation estimation. Finally, the number of traversed pixels and the raster size to obtain the final

result. The experiments show that the VLP-16 LiDAR has sparse coverage compared to the HDL-64 LiDAR and HDL-32 LiDAR but can scan fast (20 Hz) during the travel of the ROS Autonomous Navigation Vehicle and detects the geometric features of negative obstacles more clearly than the single-line LiDAR.

Moreover, since it is based on single-frame detection, this method's detection estimation efficiency is much higher than other methods. The experimental results show that the method in this paper has high reliability, realizes the function of complex LiDAR with low hardware cost, can improve the detection accuracy of negative obstacles, and has practical application value for the research of LiDAR environment perception. The experiments' performance also demonstrates the practicality of the method to meet an autonomous unmanned platform's need. It is of high value in unmanned technology, forest fire-fighting, and many other unmanned situations.

**Author Contributions:** All five authors contributed to this work. X.L. designed the research. X.L., Z.G. and X.C. processed the corresponding data. X.L. and Z.G. wrote the first draft of the manuscript. S.S. and J.L. revised and edited the final version. All authors have read and agreed to the published version of the manuscript.

**Funding:** This research was funded by the Natural Science Foundation of Heilongjiang Province of China, grant number No.LH2020C042; Fundamental Research Funds for the Central Universities, grant number No.2572019CP20.

**Institutional Review Board Statement:** Not applicable.

**Informed Consent Statement:** Not applicable.

**Data Availability Statement:** The data used to support this study's findings are available from the corresponding author upon request.

**Conflicts of Interest:** The authors declare no conflict of interest.

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
