# Peer review of "Research on Estimation Method of Geometric Features of Structured Negative Obstacle Based on Single-Frame 3D Laser Point Cloud"

_information, doi:10.3390/info12060235_

Round 1

Reviewer 1 Report

This paper introduces a method for estimating negative obstacles geometrical features in structured environments based on single-frame 3D laser point cloud. However, this manuscript is presented more like a report rather than a research paper. Several detailed critical concerns must be addressed as follows.

1. English expression needs editing and improvement. There are some grammatical mistake and typos in the manuscript, such as Page 6 Line 184-193.

2. What is the “mathematical model of the VLP-16 LiDAR negative obstacle detection range estimation” that mentioned in the Abstract?

3. The literature review is inadequate. Negative obstacle detection is a popular topic. It is necessary to provide more related works.

4. The description of the presented method is confusing. For instance, the details of PassThrough filter are not provided. How to determine the raster size and what is the influence of raster size? How to obtain Figure 3 from Figure 2? Why do you need to denoise twice in Section 2.2 and 2.6? Only straight lines can be detected and estimated in Section 3.1?

5. Can the presented method be used in unstructured environment that is more common with negative obstacles?

6. How to derive the conclusions “improve the detection accuracy of negative obstacles” and “high real-time performance”? There is no algorithm comparison and implementation time analysis in the experiment part.

Reviewer 2 Report

This paper presents a LIDAR estimation method to detect negative obstacles in structured environments.  Proposed method uses point cloud projected to a plane, StatisticalOutlierRemoval filter, median filtering and progressive probabilistic Hough transform to estimate geometrical features of the negative obstacle. Experimental results have shown high accuracy of the proposed method.
Remarks for the paper:
1. There are many grammatical and spelling errors in the text. Equations and variables in the text sometimes also appear without proper formatting (pages 10, 11, 12 and variables in the text). Question marks need to be removed when explaining equations, e.g. after equations 1, 2, 5... Overall, paper needs to be properly formatted and English language has to be checked.
2. In the experimental section, the proposed algorithm is well explained, however it should be compared with some other existing methods (e.g. that you have mentioned in the introduction). What limitations have the other methods regarding the distance from the obstacle?
3. In the experimental section, what is the overall speed of the proposed method? This should be noted, especially because it is concluded that it can work in real-time.
4. In the experimental section, what is the minimum obstacle length and width that can be detected? Probably, this minimum will also have the largest error for the maximum distance. 
5. Is it possible to use the proposed method for some other type of obstacle detection? Can you give your opinion?

Round 2

Reviewer 1 Report

Thank the authors for the efforts spent on the revision of the manuscript according my comments. However, several major concerns are still not addressed.

  1. The manuscript is still presented like a report rather than a research paper, and exists some typos (such as Page 8 Line 235). The authors should check the styles, formats and sentences throughout the manuscript.
  2. It is necessary to list the main contributions more clearly in the first part, and to focus on describing these contributions in the method part.
  3. The expression “mathematical model” is still thought to be not rigorous.
  4. Some figures are not clear, such as Figures 7, 8, 10, 11.
  5. Since the negative obstacles are more common in unstructured environments, as mentioned in the Introduction part, it is suggested to provide an experiment test in unstructured environment.

Reviewer 2 Report

The authors have addressed all of my comments.

Author Response

This manuscript is a resubmission of an earlier submission. The following is a list of the peer review reports and author responses from that submission.